# Eating Behaviours of Polish and Portuguese Adults—Cross-Sectional Surveys

**DOI:** 10.3390/nu15081934

**Published:** 2023-04-17

**Authors:** Julia Bober, Klaudia Wiśniewska, Katarzyna Okręglicka

**Affiliations:** 1Student Scientific Association of Hygiene and Prevention, Department of Social Medicine and Public Health, Medical University of Warsaw, 02-007 Warsaw, Poland; 2Department of Social Medicine and Public Health, Medical University of Warsaw, 02-007 Warsaw, Poland

**Keywords:** adults, eating behaviours, BMI, obesity, AEBQ, Poland, Portugal, nutrition education

## Abstract

Eating behaviours affect energy intake through the types and amounts of foods chosen and decisions about the beginning and ending of the eating process. This study aims to determine and compare the eating behaviours of Polish and Portuguese adults and, in addition, to establish the correlations between daily behaviours, food-approaches and food-avoidance behaviours, and BMI in both populations. The study was conducted between January 2023 and March 2023. Participants from Poland and Portugal responded to the AEBQ questionnaire and questions on eating habits and body-image self-perception. The research tool was a website-based survey questionnaire with single-choice questions. No significant differences related to BMI levels were found between the Polish and Portuguese adults in terms of their eating behaviours. Both groups were characterised by the increased intensity of their food-approach behaviours, which was directly correlated with increases in BMI. Greater snacking intensity and binge drinking were associated with higher BMI. The study revealed an increased prevalence of binge drinking in the Polish sample. The study also confirmed a higher frequency of food-approach behaviours in overweight and/or obese individuals and uncontrolled calorie intake in participants imposing dietary restrictions for weight loss. Nutrition education is needed to improve eating habits and food choices, as well as to prevent overweight and obesity in adults.

## 1. Introduction

Eating behaviours are essential daily choices, directly affecting the health and mood of individuals. The wide topic of eating behaviours consists of food choices and the motives behind them, feeding practices, dietary culture, and eating-related problems, such as eating disorders and feeding disorders [1].

Unfortunately, developed eating behaviours do not always follow the fundamentals of a healthy lifestyle. Very often, disordered eating behaviours result in health damage and may cause many diseases, including obesity, which is the biggest global public health problem [2]. For this reason, the topic of eating behaviours should be an important focus of research and intervention projects related to the etiology, prevention, and treatment of obesity and eating disorders, as well as the promotion of healthy eating patterns that might help to manage and prevent medical issues [1,3].

Over the years, increasingly concerning trends have been observed in eating behaviours. These are associated with emotional eating, stress overeating, or eating not in response to hunger, but due to a favorable environment (the unlimited sight, smell, and/or taste of palatable food). On the other hand, avoidant eating behaviours, which sometimes also result from a stress response or related to the desire for rapid weight loss, are also observed [4].

Many factors affect eating behaviours. Among the main factors are the eating behaviours of parents and other behaviours to which an individual may be exposed in their childhood. Certainly, in terms of later socioeconomic changes through life, these may be altered, but the fundamentals of the behaviours developed in childhood persist throughout life [5]. Further essential factors are personal, social, cultural, environmental, and economic [1].

The major impact of the COVID-19 pandemic on eating behaviours was observed [6,7]. The main nutrition-related behaviours that were increased were as follows: snacking with higher frequency, reaching for food/meals due to lack of activity and not hunger, and choosing foods that are sugary, highly processed, and rich in fats more often than fruits or vegetables. Furthermore, some studies have shown that the COVID-19 pandemic had a strong impact on the increased frequency and quantity of alcoholic beverage use [6]. For this reason, post-pandemic eating behaviours need special attention and education among all age groups.

With the spread of globalisation and the increased use of social media, especially among young adults, it may be argued that some environmental and cultural factors related to eating behaviours are blurred, which can cause changes in eating behaviours in various (initially dissimilar) groups. This trend can have a dual effect. On one hand, several food-related authors publish content promoting healthy lifestyles and proper nutritional choices. However, on the other hand, social media is loaded with the promotion of highly processed foods and fast foods [8,9]. Furthermore, the ubiquity of food in communication channels can result in an increased desire to procure it, as observed during the COVID-19 pandemic [10,11].

### 1.1. Body Mass Index as a Health-Related Indicator

The body mass index (BMI) is a widely applicable, fast, and affordable tool, essential for assessing information on excessive body weight. Essentially, although it is calculated for individuals, it does not take age, gender, or ethnicity into consideration, with the effect that the results obtained should be interpreted in general terms. As BMI does not measure excess body-fat mass but focuses on total body weight, certain individuals with high, lean body mass who are not overweight and/or obese may have increased BMI values [12,13].

The current World Health Organisation (WHO) classification of BMI highlights six stages:Underweight, BMI score < 18.5 kg/m^2^;Normal weight (optimal), 18.5–24.9 kg/m^2^;Overweight (pre-obese), 25–29.9 kg/m^2^;Obese type I ≥ 30–34.9 kg/m^2^;Obese type II 35–39.9 kg/m^2^;Obese type III ≥ 40 kg/m^2^ [13].

The use of age-standardised BMI for adults in Poland and Portugal shows an increase in mean BMI results over the years for both sexes in both countries. Higher BMI scores were observed in Poland, equally for both genders and for women and men separately. In the latest data, from 2016, the age-standardised estimates for adults in Poland amounted to 26.7, while in Portugal, the score was 25.6 [14]. It follows that in both countries, adults frequently present abnormal BMI levels, indicating excess body weight. Changes in eating behaviours have exerted a major impact on the increase in average BMI levels over the years.

### 1.2. Poland and Portugal, Nutritional- and Health-Status Characteristics

The dietary recommendations in Portugal and Poland are based on the latest scientific reports and, despite some differences, they are mostly in line with the WHO recommendations [15,16]. Both countries significantly highlight energy requirements, the intake of protein, carbohydrates, including fiber, and fats, as well as micro- and macro-nutrients. Furthermore, special attention is paid to the intake of dietary sodium, the excessive consumption of which directly affects the prevalence of cardiovascular diseases, which are among the major health-care issues in both countries [17,18]. Despite their completely different cultures, geographic locations, and food industries, Portugal and Poland seem to face comparable problems in terms of proper nutrition and public health.

### 1.3. Portugal

Although it is considered by some to be a Mediterranean diet, this is not entirely true of the diet of Portuguese adults. Since the late 1990s, a continuous decline in the use of the fundamentals of the Mediterranean diet has been observed among the Portuguese people which, to be clear, was never high [19,20]. Furthermore, a decrease in the energy supplied from products classified as Mediterranean was observed [19,20]. The latest dietary requirements for the Portuguese population are illustrated in the Portuguese Food Wheel Guide. However, the low levels of adherence to these guidelines are concerning. The consumption of food groups labeled as “meat, fish, and eggs,” as well as “dairy,” was higher than the recommended levels, while the consumption of “fruit” and “vegetables” was below the values recommended in the guidelines. Notably, the consumption of seafood and fish by most Portuguese adults matches the guidelines [16].

Inadequate nutrition is one of the causes of chronic non-transmissible diseases, including obesity, oncological diseases, cardiovascular diseases, and type 2 diabetes. In Portugal, in 2019, improper nutrition contributed to 11.4% of the total mortality and overweight and/or obesity, which affect more than half of the Portuguese population, contributing to about 9% of the mortality rate [16].

The major unhealthy food-related behaviours of the Portuguese population include the high consumption of red and processed meat, the low consumption of whole grains, and high sodium intake. Despite the wide availability of vegetables and fruits, their daily consumption is below the recommended levels. All these behaviours reduce lifespan and health quality. Furthermore, the diet of Portuguese people is characterised by an increased percentage, relative to the guidelines, of free sugars, and a high percentage of highly processed foods and snacks, both salty and sweet [17].

More research is still needed to determine the total impact of the COVID-19 pandemic on food consumption and nutritional status of the Portuguese population, but the short-term results show a negative impact on food and nutritional status. The main differences are in the higher snacking intensity and the total number of calories consumed per day [17]. No research investigating eating behaviours of the Portuguese adult population following the COVID-19 pandemic has yet been performed.

Despite the existence of focused nutrition programs for the Portuguese population, slight improvements in healthy eating and decreased overweight prevalence are observed only among children and adolescents [17,21,22]. The reason for this may be the fact that most of the programs are targeted at this age group [17]. This indicates the need to include adults in education programs regarding the prevention of nutrition-related, chronic, non-transmissible diseases.

The average daily calorie supply available for consumption per person is significantly high, between 3.480 (data from 2018) and 4.075 (data from 2016–2020, including first period of the pandemic) calories, which is twice the recommended value of daily energy intake for an adult of average, healthy weight [23,24,25].

### 1.4. Poland

Poland, considered a developing country, is currently facing major public health and health-care problems, partially associated with excessive body weight [18]. The latest dietary guidelines were intended to help promote healthy eating habits in the community. Unfortunately, despite online and free access, few people are aware of the existence of these guidelines, while an equally large group of people do not know how to use them.

A new report on the eating behaviours of Poles after the COVID-19 pandemic showed a lower adherence to the WHO and Polish guidelines; even before 2019, they were not particularly high. Furthermore, adherence to the Mediterranean diet, which is recommended and whose positive effects on health have been scientifically confirmed, was lower, showing only a slight correlation [18]. An increased intake of calories was observed, as well as the consumption of red meat, processed meat, table salt, fast food, and alcohol [18,26]. On the other hand, a slight increase in the consumption of vegetables and fruit was observed, although the level was still below that recommended by the WHO [15,18].

Due to daily eating behaviours, an increase in the intake of total fat, animal fat, protein, and carbohydrates was observed. The increased calorie and fat intake was especially alarming, since it was already excessive before the pandemic. Furthermore, fiber intake, which usually should be as high as possible (in healthy adults), was significantly lower than the recommended values. Fish consumption in Poland remains at a low level, and most people do not follow the recommendation to consume fish twice per week. This might be one of the causes of the low values of omega-3 fatty acids in the diet of Poles [15].

During recent years, the Polish population began to consume sweets more frequently, which significantly increased calorie intake. Regarding meal frequency, most Poles were observed to increase the number of meals during the pandemic to five or more. Snacking frequency also increased, including the consumption of sweets and salty snacks, but also of fruits. These behaviours also increased caloric intake and, ultimately, led to body-mass changes and increases in BMI levels. Notably, almost half of Poles increased their body weight during the pandemic [18].

Most Poles did not follow a specific dietary pattern, including the Mediterranean diet. A very small percentage declared that they followed a vegetarian or vegan diet. Furthermore, the Polish population were observed to have difficulties with healthy eating, usually due to a lack of interest or the high prices of food products [18]. An alarming problem that could become a concern in the future is the low interest of Poles in healthy eating, along with the impact of nutrition on health. Most of the knowledge cited by the population is derived from the internet, social media, and friends, and not, as it should be, from a specialist [8,18].

In Poland, the average daily calorie supply available for consumption in 2018 (before the COVID-19 pandemic) oscillated around 3.537 calories per day per person. More recent, post-pandemic data are not available, but the value of this figure can be expected to increase based on data from other countries and previous studies [24].

### 1.5. Similarities in Nutrition and Health Status in Portugal and Poland

Both countries have a similar share of adults who are obese (23.2% in Portugal, 25.6% in Poland). In both, more than 50% of adults are overweight or obese (57.5% in Portugal, 58.3% in Poland), which is an alarming trend that should stimulate action to reduce individuals’ body weights to normal values [27]. Regardless of the differences in terms of food markets and the availability of certain products, the Portuguese and Polish populations seem to make the same errors in terms of food choices and proper nutrition. This results in the widespread prevalence of chronic, non-transmissible diseases and an increase in mortality.

The high calorie intake in both countries, as well as the widespread prevalence of overweight and obesity, may be due not only to improper food choices and the failure to follow the guidelines for the main food groups, but also to an increased incidence of food-approach behaviours, as well as a lower incidence of food-avoidance behaviours.

The aim of this study was to analyse and compare the eating behaviours of Polish and Portuguese adults and, in addition, to establish correlations between daily habits, food-approach, food-avoidance behaviours, and BMI in both groups.

## 2. Materials and Methods

The participants (N = 531, Npl = 225, Npt = 306), recruited between January 2023 and March 2023 via online social network sites, completed an online version of the AEBQ (Adult Eating Behaviour Questionnaire). Validated AEBQ was performed for specific language versions, both Polish and Portuguese [3,28]. The survey was carried out in Poland and Portugal. In total, 172 women in Polish and 229 in Portuguese sample participated in the study. In terms of men, 53 from Poland and 72 from Portugal completed the survey. In addition to the AEBQ questionnaire, participants self-reported age, gender, anthropometric data, and ethnic background. Individuals also answered questions about perception of current body image and daily behaviours in relation to consumed meals. This part of the questionnaire was based on self-administered questions designed using the Food Frequency Questionnaire (FFQ), which was validated in a previous study on a Portuguese sample [29]. The self-reported questions were next validated with a group of 15 Poles and 15 Portuguese to ensure understanding of the questions. From the reported weight (kg) and height (m), BMI was calculated as weight (kg)/height^2^ (m^2^). Two language versions (Polish, Portuguese) of the survey were used to prevent misunderstanding of questions. Questions in both surveys were the same, in the same order.

The exclusion criteria were age under 18 years old and t nationality other than Polish or Portuguese. The inclusion criteria were conscious consent to participate in an anonymous survey, agreement to further use of the results in scientific work, completion of all questions in the survey, Portuguese or Polish nationality, and age over 18 years old. Initially, 572 respondents that fitted the inclusion criteria, and 537 were left after exclusion. In the final analysis, 531 individuals were included, as the rest failed to answer all the required questions in the survey (Figure 1).

The collected data were subjected to statistical analysis. Original, 8-factor AEBQ was conducted due to good reliability in all scales. In order to verify whether the numerical distributions obtained from the respondents’ answers differed from the hypothetical normal distribution, Kolmogorov–Smirnov tests, recommended for situations with relatively large sample sizes, were calculated [30].

The 35-item AEBQ features four food-approach subscales, hunger (5 items), food responsiveness (4 items), emotional overeating (5 items), and enjoyment of food (3 items), and four ‘food avoidance’ subscales, satiety responsiveness (3 items), emotional under-eating (5 items), food fussiness (5 items), and slowness in eating (4 items). Responses from each item were recorded on a 5-point Likert scale ranging from ‘Strongly Disagree’ to ‘Strongly Agree’. Average scores were calculated for each subscale. To determine the level of reliability, understood as the accuracy of the measurement of the questionnaire used, the Cronbach’s alpha statistic was calculated both for the entire study group and by nationality.

The correlation analysis, using quantitative data with significantly different distributions from the normal range, was carried out using Spearman’s nonparametric rho correlation test based on ranks [30,31]. Furthermore, for ordinal variables, Kendall’s nonparametric Tau-b correlation test was used, as it allows the analysis of scales with small ranges [30,31].

Comparisons between the two groups were performed using the Mann–Whitney U test. To assess the strengths of the effects, Glass’ bivariate correlation coefficient (rg = 0.40) was used and presented in the tables. The epsilon square factor (ε2) was used to evaluate the effect size and is presented in the tables. Post hoc tests were calculated using the Bonferroni–Dunn method. To test the influence of individual demographic variables on the dependent variables, an analysis of between-group differences was performed in a two-factor ANOVA with covariates.

## 3. Results

### 3.1. Study Group

The study included 531 individuals of Polish and Portuguese nationality, between the ages of 18 and 78, including both men and women. Their body weights were between 40 and 124 kg, and their heights were between 145 and 198 cm (Table 1).

The study’s findings also showed that compared to the Portuguese, the Poles were older, taller, and heavier. Detailed descriptive statistics for the described sample can be found in Table 2.

### 3.2. Health-Related Measurements

The impact of the demographic variables of gender, age, and nationality on the BMI levels showed a significant effect for gender (*p* = 0.001) and age (*p* = 0.026), confirming the presence of higher BMI levels among the men than among the women, as well as higher BMI levels among the older adults. Notably, no significant relationship was found between nationality and BMI. The exact outcomes can be found in Table 3.

The coefficients indicated that BMI increases with age (*p* = 0.0066) and intensity of binge drinking (*p* = 0.0074). The study found no significant changes in BMI or in variables such as the number of meals, number of skipped meals, or time (hour) of consuming meals. The results of this analysis are presented in Table 4.

Furthermore, it was found that the vegetarians, vegans, and flexitarians had significantly lower BMI levels than those who consumed all products (*p* = 0.016). Detailed results of the above analysis can be found in Table 5.

### 3.3. Daily Behaviours

The analyses showed that the Polish participants were more likely to eat a second breakfast or morning snack over the course of a week, while the Portuguese were more likely to eat lunch, dessert or an afternoon snack, and dinner. In addition, the Poles were more likely to eat a second breakfast or morning snack and lunch later in the day, while the Portuguese were more likely to eat dessert or a snack later in the night. Moreover, the Poles were more likely than the Portuguese to skip meals and over-consume alcohol. Detailed results of this analysis can be found in Table 6.

### 3.4. Eating Behaviours

The level of reliability for the eight-factor AEBQ questionnaire was good or very good in most of the dimensions of the AEBQ questionnaire, ranging from 0.66, for the hunger subscale, to 0.89, for emotional overeating. The level of reliability of the survey instrument used did not differ when the groups were divided by nationality compared to the entire sample. Detailed results of this analysis can be found in Table 7.

An analysis of the cross-group comparisons of the responses to the AEBQ questionnaire showed that compared to the Portuguese, the Poles had slightly higher scores for the enjoyment of food and emotional undereating. In contrast, the Portuguese had higher scores on the scales of emotional overeating and food fussiness. The remaining results, as well as the scores for the food-approach and food-avoidance scales, showed no significant differences. For this reason, the subsequent analysis considered the group as one, with no nationality distinction. Detailed results of this analysis are provided in Table 8.

Furthermore, as the highest-scoring dimension of the AEBQ questionnaire, food approach, increased, the weekly frequency of eating desserts or late-night snacks, binge drinking, and BMI levels increased, and the number of skipped meals decreased (*p* = 0.0082, *p* = 0.0069, *p* = 0.0073, and *p* = 0.0079, respectively). Moreover, as the second-highest-scoring dimension of the AEBQ questionnaire, food avoidance, increased, the BMI level decreased, as well as the weekly frequency of eating lunch and binge drinking (*p* = 0.0059, *p* = 0.0064, and *p* = 0.0078, respectively). The results are shown in Table 9.

Another analysis showed statistically significant differences between the groups according to BMI level. Differences were observed in the dimensions of emotional overeating, food avoidance, emotional undereating, and slowness in eating. It was found that the individuals who had an optimal BMI were characterised by significantly lower scores on the food-approach dimension than the overweight individuals. Moreover, those with optimal or low BMI levels had significantly lower scores on emotional overeating than those who were overweight. The overweight individuals had significantly lower scores for food avoidance, emotional undereating, and slowness in eating than the underweight individuals or those with optimal BMI levels. Detailed results of this analysis can be found in Table 10.

Furthermore, an analysis of differences between the AEBQ scores in relation to perceptions of body image and weight showed statistically significant differences for the following variables: food approach (*p* = 0.001), hunger (*p* = 0.048), food responsiveness (*p* = 0.039), emotional overeating (*p* = 0.001), food avoidance (*p* = 0.012), emotional undereating (*p* = 0.003), food fussiness (*p* = 0.006), and slowness in eating (*p* = 0.031). It was found that the individuals with neutral perceptions about their weight scored significantly lower than those who rated their own weight as negative on food approach and emotional overeating. In contrast, the individuals with positive perceptions of their body image and weight scored significantly lower on food approach, hunger, food responsiveness, and emotional overeating than those with negative perceptions about their body image and weight.

Moreover, the individuals who had positive perceptions about their body image and weight also tended to have lower food-avoidance and food-fussiness scores than those with negative perceptions. The individuals with negative perceptions about their body image and weight had lower scores on food-avoidance and emotional undereating than those with neutral perceptions, and lower scores on slowness in eating than the individuals with positive perceptions, as shown in Table 11.

A further analysis focused on the desire to change body image and/or weight showed statistically significant differences for the dimensions of food approach (*p* = 0.006), emotional overeating (*p* = 0.001), food avoidance (*p* = 0.001), emotional undereating (*p* = 0.001), and slowness in eating (*p* = 0.004).

It was found that the individuals who wanted to change the appearance of their body shape, but did not experience significant weight changes, obtained lower scores on the food approach and emotional overeating scales than those who wanted to reduce their weight, and lower scores on emotional undereating than those who wanted to increase their weight. In contrast, the individuals who wanted to increase their body weight had significantly lower levels of emotional overeating than those wanting to lose weight.

Furthermore, the respondents who wanted to reduce their body weight had significantly lower scores on food avoidance compared to those who wanted to change their body shape without changing their weight, or who wanted to increase their weight. In addition, it was noted that the group of participants who wanted to reduce their weight had lower scores on emotional undereating and slowness in eating than those who wanted to gain weight, as shown in Table 12.

The results of the analysis of the effect of the type of dietary pattern on the various dimensions of the AEBQ questionnaire did not reveal any statistically significant differences. Therefore, the data from this analysis were withdrawn from publication.

## 4. Discussion

To the best of our knowledge, this is the first study to compare eating behaviours in Poland and Portugal, two countries with completely different food and eating cultures. The present study showed that there are no noticeable differences between the BMI levels of the two nationalities. This supports previous results that overweight is increasing regardless of differences in food culture, in both developing and developed countries [32,33]. Although the study showed normal weights in most of the groups, the problem of overweight and/or obesity was nevertheless present in this sample. One in four individuals were overweight or obese and, considering that most of the sample consisted of young adults, and that BMI increases with age, the number of overweight and/or obese people, over a number of years, may be increasing [12,34,35]. The prevention of overweight and obesity should be a focus of the healthcare systems of both countries [25,36,37].

Differences in eating culture due to geographic location, as well as lifestyle habits, were associated with the more frequent consumption of meals in the first part of the day in Poland, while in Portugal, more meals were consumed in the afternoon and evening. These differences, however, showed no correlation with BMI. Although some sources link the consumption of higher numbers of calories in the second half of the day with excessive weight gain, no confirmation was found in this case [38,39,40,41]. This may have been related to the fact that in both Poland and Portugal, most individuals significantly exceed their caloric requirements each day [17,18,24]. A major concern is binge drinking, which is the frequent consumption of alcohol in large quantities. This occurrence was observed with greater intensity in Poland than in Portugal, which may explain the fact that despite their higher number of skipped meals, the Poles had similar BMI values. This might be linked to the high caloric content of alcohol consumed [42,43,44].

This study confirmed that the participants following plant-based diets and/or limiting their meat intake had lower BMI [45,46,47]. Moreover, the eating behaviours of these individuals, as measured by the AEBQ, did not differ from those of the omnivores, confirming that people on plant-based diets consume fewer calories due to the products they eat, rather than their eating behaviour.

Food-approach behaviours were found to be more frequent than food-avoidance behaviours for both groups. Therefore, no nationality differences were noticed. This was consistent with previous studies on other developed countries [48,49,50,51]. Food-approach behaviours are linked with the consumption of a higher number of meals/snacks during the day, including additional meals, such as late-night snacks. These results were confirmed in this study and in others [38,39]. A higher incidence of binge drinking has also been linked with behaviours that increase food intake. All these factors influenced the increased calorie intake in both countries. Moreover, the BMI levels were significantly higher in the individuals with higher incidences of food-approach behaviours than food-avoidance behaviours, which was also shown in previous studies [48,49,50,52,53,54,55,56].

Nutrition education is needed, especially for individuals with overweight/obesity, as they engage more frequently in behaviours such as the overconsumption of food under the influence of emotions [56,57]. The education provided should not only concern proper nutrition, but also how to manage emotions and stress [57,58]. According to previous research, the use of mindfulness techniques can be found helpful [59,60].

Due to the unrealistic body-shape ideals that are heavily promoted through social media, it is difficult, especially for young adults, to maintain a positive attitude toward body image [9,61,62]. Negative perceptions about body image are often associated with disordered eating behaviours, and may result in eating disorders, which are usually maintained by food-avoidance behaviours [63,64,65]. A higher intensity of these behaviours in individuals with negative perceptions was also observed in this study. Positive and/or neutral perceptions of body image/weight were not related to alarming or extreme eating behaviours, which can lead to the development of eating disorders.

Intriguing results were obtained regarding the desire for weight change and the various dimensions of the AEBQ questionnaire. These results may confirm previous studies’ findings that excessive dietary restrictions, associated with the desire to lose weight, intensify food-approach behaviours that result in snacking, as well as the desire to overeat [66,67,68]. This is another reason for the focus on nutrition-education programs, including proper eating behaviours, in both countries.

## 5. Conclusions

This study showed no significant differences in eating behaviours that can affect to BMI levels between Polish and Portuguese adults. However, both groups were characterised by the increased intensity of their food-approach behaviours, which were directly correlate with increases in BMI. Other daily behaviours related to the distribution over the day and/or the timing of meals consumed, by country, did not affect the differences between the participants’ BMI levels. Nevertheless, greater snacking intensity, including late-night snacks, as well as binge drinking, were associated with higher BMI.

In conclusion, although some research links the consumption of higher numbers of calories in the second half of the day with excessive weight gain, no confirmation was found in this study, which can be helpful for creating dietary plans for patients with obesity and metabolic disorders with different meal-time preferences.

Overall, we conclude that plant-based diets can result in beneficial body weight in the young population, even without dietary caloric restriction.

In addition, the study revealed an increased prevalence of binge drinking in the Polish sample. The study also confirmed a higher frequency of food-approach behaviours in overweight and/or obese individuals and uncontrolled calorie intake in participants imposing dietary restrictions for weight loss. Health education, focused on proper nutrition and lifestyles, is needed in both Poland and Portugal. This should be a focus for health-care systems in upcoming years to prevent the overweight-and-obesity pandemic not only in children, but also in adult populations. More research is needed to establish correlations between eating behaviours and different daily behaviours in various groups to obtain a wider perspective on this phenomenon.

## Figures and Tables

**Figure 1 nutrients-15-01934-f001:**
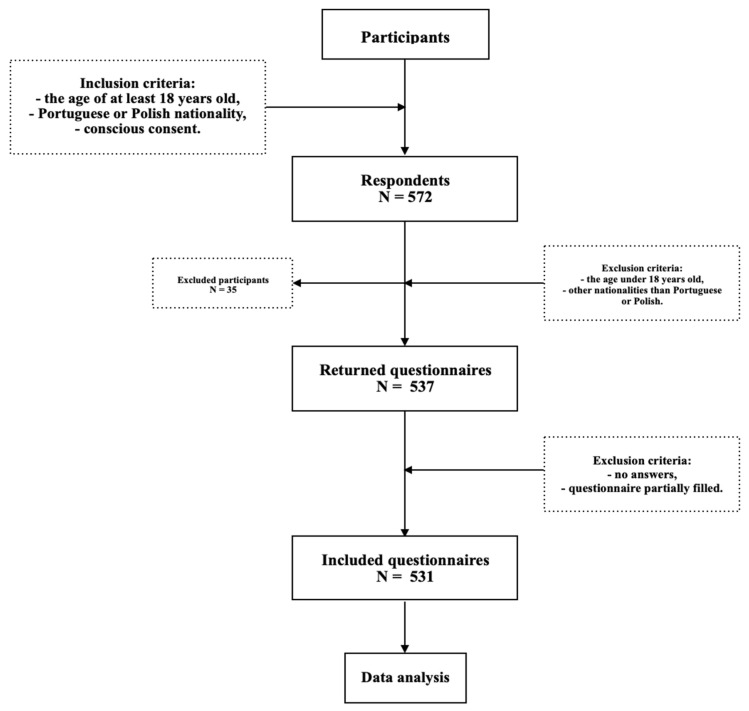
Flowchart: study design and data collection (N—number of participants).

**Table 1 nutrients-15-01934-t001:** Descriptive statistics for demographic variables of the sample (N = 531).

	R	M	SD	Mdn	Sk	Kurt	D
total							
age	18–78	25.84	8.91	23	2.09	5.13	0.23 **
weight	40–124	64.75	13.37	62	1.11	1.87	0.10 **
height	145–198	168.39	8.99	168	0.53	0.11	0.09 **
Portuguese							
age	18–66	23.79	7.40	22	2.51	7.45	0.24 **
weight	40–115	62.74	11.84	60	0.86	0.90	0.11 **
height	145–192	166.58	8.35	165	0.46	−0.16	0.11 **
BMI	15.06–34.72	22.52	3.33	22.04	0.91	1.09	0.08 **
Polish							
age	18–78	28.62	9.98	24	1.74	3.54	0.21 **
weight	43–124	67.49	14.80	64	1.14	1.74	0.11 **
height	151–198	170.86	9.26	170	0.54	0.11	0.07 **
BMI	15.32–41.52	22.99	3.99	22.49	1.52	4.19	0.09 **
men							
age	18.00–78.00	27.56	10.63	23.00	2.04	4.80	0.20 **
weight	48.00–121.00	77.29	13.20	77.00	0.55	1.04	0.10 **
height	164.00–198.00	179.04	7.13	179.00	0.41	−0.07	0.08
BMI	15.32–36.93	24.03	3.37	23.91	0.62	1.58	0.07
women							
age	18.00–71.00	25.32	8.29	23.00	1.99	4.29	0.24 **
weight	40.00–124.00	60.82	10.82	59.00	1.52	5.01	0.10 **
height	145.00–183.00	165.07	6.67	164.00	0.22	−0.17	0.08 **
BMI	15.06–41.52	22.30	3.62	21.71	1.61	4.53	0.10 **

** *p* < 0.01. R: range, M: mean, SD: standard deviation, Mdn: median, Sk: skewness of distribution, Kurt: Kurtosis, D: Kolmogorov–Smirnov-test result.

**Table 2 nutrients-15-01934-t002:** Results of analysis using Mann–Whitney U test on demographic variables by nationality.

Variables	Portuguese (*n* = 306)		Polish (*n* = 225)				
	Mdn	Mrang	Mdn	Mrang	U	*p*	Rg
age	22.00	220.49	24.00	327.90	20498.00	<0.001	0.40
weight	60.00	245.52	64.00	293.86	28157.00	<0.001	0.18
height	165.00	235.95	170.00	306.86	25230.50	<0.001	0.27
BMI	22.04	259.78	22.49	274.46	32522.50	0.276	0.06

Mdn: median, Mrang: mean rang, U: Mann–Whitney U test, *p*: *p*-value, Rg: strength measurement, Glass’ bivariate correlation coefficient.

**Table 3 nutrients-15-01934-t003:** Impact of demographic variables on BMI levels.

Source of Variation	*F*	*p*	η^2^
nationality	0.63	0.427	0.00
gender	4.96	0.026	0.01
age	21.94	<0.001	0.04
nationality × gender	0.05	0.818	0.00
nationality × age	1.25	0.265	0.00
gender × age	0.64	0.423	0.00
nationality × gender × age	0.03	0.857	0.00

*F*: ANOVA test, *p*: *p*-value, η^2^: strength of the eta^2^ effect.

**Table 4 nutrients-15-01934-t004:** Kendall’s tau-b correlation scores between variables and BMI levels (N = 531).

Variable	BMI
age	0.16 **
weekly frequency: breakfast	−0.01
weekly frequency: second breakfast/morning snack	−0.02
weekly frequency: lunch	−0.04
weekly frequency: dessert/afternoon snack	−0.05
weekly frequency: dinner	−0.01
weekly frequency: dessert/night snack	0.00
time (hour): breakfast	0.03
time (hour): second breakfast/morning snack	0.01
time (hour): lunch	−0.01
time (hour): dessert/afternoon snack	−0.03
time (hour): dinner	0.00
time (hour): dessert/night snack	−0.03
number of skipped meals	0.02
binge drinking	0.14 **

** *p* < 0.01.

**Table 5 nutrients-15-01934-t005:** Results of intergroup comparison using Kruskal–Wallis H test, according to BMI level and dietary pattern.

	Group	Mdn	Mrang	H (2)	*p*	Post-Hoc
	Dietary pattern					
I	omnivore	22.41	280.00	10.26 *	0.016	II < I *
II	vegetarian/vegan	21.10	225.51			III < I *
III	flexitarian	21.84	235.38			

* *p* < 0.05. Mdn: median, Mrang: mean rang, H (2): H Kruskal–Wallis test, *p*: *p*-value.

**Table 6 nutrients-15-01934-t006:** Results of analysis using Mann–Whitney U test for study indicators by nationality.

Variables	Portuguese (*n* = 298)		Polish (*n* = 222)				
	Mdn	Mrang	Mdn	Mrang	U	*p*	Rg
weekly frequency: breakfast	4	255.38	4	267.37	31,552.50	0.258	0.05
weekly frequency: second breakfast/morning snack	2	233.91	2	305.02	24,748.00	<0.001	0.27
weekly frequency: lunch	4	289.24	4	226.62	25,547.00	<0.001	0.24
Weekly frequency: dessert/afternoon snack	3	311.93	2	200.13	19,628.50	<0.001	0.42
weekly frequency: dinner	4	286.33	4	237.26	27,958.50	<0.001	0.19
weekly frequency: dessert/night snack	1	262.82	1	264.43	33,577.00	0.898	0.01
time (hour): breakfast	8	233.85	8	248.58	26,183.50	0.237	0.06
time (hour): second breakfast/morning snack	11	171.59	11	203.84	14,217.50	0.002	0.17
time (hour): lunch	13	195.42	14	335.49	13,369.50	<0.001	0.56
time (hour): dessert/afternoon snack	17	230.63	17	224.71	23,322.50	0.632	0.03
time (hour): dinner	20	299.39	20	210.71	22,064.50	<0.001	0.34
time (hour): dessert/night. snack	23	181.05	21	89.38	3582.50	<0.001	0.64
number of skipped meals	1	251.61	1	284.48	30,020.50	0.010	0.12
binge drinking	0	253.13	1	278.94	30,539.00	0.038	0.10

Mdn: median, Mrang: mean rang, U: Mann–Whitney U test, *p*: *p*-value, Rg: strength measure, Glass’ bivariate correlation coefficient.

**Table 7 nutrients-15-01934-t007:** Descriptive statistics for the dimensions of the AEBQ questionnaire, along with Cronbach’s alpha statistics in each group.

	Portuguese			Polish			All Sample		
	M	SD	α	M	SD	α	M	SD	α
food approach	2.88	0.60	0.84	2.89	0.62	0.86	2.88	0.60	0.84
hunger	2.58	0.71	0.65	2.66	0.76	0.69	2.62	0.73	0.66
food responsiveness	2.66	0.76	0.69	2.68	0.79	0.72	2.67	0.77	0.70
emotional overeating	2.85	1.08	0.89	2.63	1.05	0.88	2.75	1.07	0.89
enjoyment of food	3.73	0.75	0.75	3.95	0.95	0.92	3.82	0.85	0.78
food avoidance	2.52	0.59	0.82	2.55	0.54	0.82	2.53	0.57	0.82
satiety responsiveness	2.47	0.74	0.66	2.47	0.75	0.68	2.47	0.74	0.67
emotional undereating	2.56	1.03	0.89	2.82	1.01	0.86	2.67	1.03	0.88
food fussiness	2.37	0.98	0.89	2.16	0.86	0.85	2.28	0.94	0.88
slowness in eating	2.70	1.08	0.88	2.78	1.00	0.85	2.73	1.05	0.87

M: mean, SD: standard deviation, α: α-Cronbach reliability measure.

**Table 8 nutrients-15-01934-t008:** Results of Mann–Whitney U test on AEBQ-questionnaire dimensions according to nationality.

Variables	Portuguese (*n* = 306)		Polish (*n* = 225)				
	Mdn	Mrang	Mdn	Mrang	U	*p*	rg
food approach	2.88	266.14	2.88	265.81	34,381.50	0.980	0.00
hunger	2.60	261.32	2.60	272.36	32,994.00	0.411	0.04
food responsiveness	2.50	264.77	2.75	267.68	34,047.50	0.828	0.01
emotional overeating	2.80	280.65	2.40	246.08	29,942.00	0.010	0.13
enjoyment of food	3.67	242.56	4.00	296.83	27,253.50	<0.001	0.20
food avoidance	2.50	261.29	2.56	272.41	32,983.00	0.409	0.04
satiety responsiveness	2.50	266.97	2.50	264.68	34,129.00	0.865	0.01
emotional undereating	2.55	249.38	2.80	288.60	29,339.00	0.004	0.15
food fussiness	2.20	279.35	2.00	247.84	30,339.00	0.019	0.12
slowness in eating	2.50	259.98	2.75	274.19	32,581.50	0.290	0.05

Mdn: median, Mrang: mean rang, U: Mann–Whitney U test, *p*: *p*-value, Rg: strength measure, Glass’ bivariate correlation coefficient.

**Table 9 nutrients-15-01934-t009:** Relationships between variables—Kendall’s tau-b correlation coefficients (N = 531).

Variables	Food Approach	Food Avoidance
age	−0.02	−0.04
weekly frequency: breakfast	0.05	−0.03
weekly frequency: second breakfast/morning snack	0.02	−0.03
weekly frequency: lunch	0.02	−0.10 **
weekly frequency: dessert/afternoon snack	0.03	−0.04
weekly frequency: dinner	−0.00	−0.06
weekly frequency: dessert/night snack	0.13 **	0.01
time (hour): breakfast	0.02	0.03
time (hour): second breakfast/morning snack	0.01	0.07
time (hour): lunch	−0.01	0.03
time (hour): dessert/afternoon breakfast	−0.01	0.06
time (hour): dinner	−0.01	0.03
time (hour): dessert/night snack	−0.05	0.01
number of skipped meals	−0.12 **	−0.00
binge drinking	0.11 **	−0.10 **
BMI	0.09 **	−0.19 **

** *p* < 0.01. Kendall’s tau-b correlation coefficient.

**Table 10 nutrients-15-01934-t010:** Results of between-group comparison using Kruskal–Wallis H test, along with post hoc tests on AEBQ-questionnaire dimensions and BMI levels.

		I: Underweight (*n* = 32)		II: Normal Weight (*n* = 303)		III: Overweight (*n* = 173)					
		Mdn	Mrang	Mdn	Mrang	Mdn	Mrang	H (2)	*p*	ε^2^	Post-Hoc
A	food approach	2.76	228.00	2.82	236.43	3.00	291.05	16.37	<0.001	0.03	A.II < A.III **
B	hunger	2.90	301.81	2.60	247.28	2.60	258.40	4.21	0.122	0.01	n.i.
C	food responsiveness	2.50	229.95	2.50	246.09	2.75	273.77	4.92	0.085	0.01	n.i.
D	emotional overeating	2.23	209.97	2.40	226.48	3.20	311.81	40.53	<0.001	0.08	D.I < D.III ** D.II < D.III **
E	enjoyment of food	3.67	217.91	4.00	262.77	3.67	245.27	3.71	0.157	0.01	n.i.
F	food avoidance	2.81	305.56	2.56	265.11	2.44	226.48	11.77	0.003	0.02	F.III < F.I * F.III < F.II *
G	satiety responsiveness	2.50	262.56	2.50	250.39	2.50	260.20	0.60	0.740	0.00	n.i.
H	emotional undereating	3.00	307.91	2.80	264.49	2.20	227.13	11.72	0.003	0.02	H.III < H.I * H.III < H.II *
I	food fussiness	2.40	269.63	2.00	248.32	2.20	262.52	1.40	0.496	0.00	n.i.
J	slowness in eating	3.00	296.44	2.75	272.14	2.25	215.85	19.08	<0.001	0.04	J.III < J.I * J.III < J.II **

* *p* < 0.05, ** *p* < 0.01. Mdn: median, Mrang: mean range, H (2): H Kruskal–Wallis test, *p*: *p*-value, ε^2^: measurement of the strength of the effect by the epsilon2 coefficient.

**Table 11 nutrients-15-01934-t011:** Results of intergroup comparison using Kruskal–Wallis H test, along with post hoc tests on AEBQ-questionnaire dimensions and types of perception of body image and weight.

Variables		I: Negative (*n* = 172)		II: Neutral (*n* = 183)		III: Positive (*n* = 169)					
		Mdn	Mrang	Mdn	Mrang	Mdn	Mrang	H (2)	*p*	ε^2^	Post-Hoc
A	food approach	3.06	306.87	2.88	250.78	2.71	230.03	23.65	<0.001	0.05	A.II < A.I ** A.III < A.I **
B	hunger	2.60	280.03	2.60	266.42	2.40	240.42	6.07	0.048	0.01	B.III < B.I *
C	food responsiveness	2.75	285.34	2.50	257.35	2.50	244.83	6.49	0.039	0.01	C.III < C.I *
D	emotional overeating	3.20	323.30	2.40	236.69	2.40	228.57	41.72	<0.001	0.08	D.II < D.I ** D.III < D.I **
E	enjoyment of food	3.67	252.71	3.67	258.02	4.00	275.71	2.21	0.332	0.00	n.i.
F	food avoidance	2.44	247.08	2.61	289.20	2.50	249.28	8.77	0.012	0.02	F.I < F.II * F.III < F.II *
G	satiety responsiveness	2.50	276.02	2.50	267.80	2.25	243.00	4.45	0.108	0.01	n.i.
H	emotional undereating	2.40	236.49	2.80	290.49	2.60	258.66	11.50	0.003	0.02	H.I < H.II **
I	food fussiness	2.20	273.77	2.20	280.05	2.00	232.03	10.32	0.006	0.02	I.III < I.I * I.III < I.II **
J	slowness in eating	2.50	238.40	2.67	269.34	2.75	279.62	6.93	0.031	0.01	J.I < J.III *

* *p* < 0.05, ** *p* < 0.01. Mdn: median, Mrang: mean range, H (2): H Kruskal–Wallis test, *p*: *p*-value, ε^2^: measurement of the strength of the effect by the epsilon2 coefficient.

**Table 12 nutrients-15-01934-t012:** Findings of intergroup comparison using Kruskal–Wallis H test along with post hoc tests on AEBQ-questionnaire dimensions and desire to change body image and/or weight.

Variables		I: No (*n* = 76)		II: Yes, Reduce Body Weight (*n* = 198)		III: Yes, Change the Appearance of the Body Shape, without Significantly Changing the Body Weight (*n* = 202)		IV: Yes, Gain Body Weight (*n* = 43)					
		Mdn	Mrang	Mdn	Mrang	Mdn	Mrang	Mdn	Mrang	H (3)	*p*	ε^2^	Post-Hoc
A	food approach	2.71	236.27	3.00	289.57	2.82	243.39	2.81	243.84	12.59	0.006	0.02	A.III < A.II *
B	hunger	2.60	268.33	2.60	250.00	2.60	258.50	2.60	298.36	3.98	0.264	0.01	n.i.
C	food responsiveness	2.50	236.25	2.75	273.30	2.50	256.40	2.50	257.66	3.63	0.305	0.01	n.i.
D	emotional overeating	2.40	231.84	3.20	309.54	2.60	233.25	2.20	207.31	36.17	<0.001	0.07	D.I < D.II ** D.III < D.II ** D.IV < D.II **
E	enjoyment of food	4.00	255.40	4.00	263.02	4.00	254.82	4.00	272.62	0.71	0.872	0.00	n.i.
F	food avoidance	2.56	273.32	2.43	229.40	2.61	273.97	2.72	311.71	15.71	0.001	0.03	F.II < F.III * F.II < F.IV **
G	satiety responsiveness	2.50	263.23	2.50	254.49	2.50	270.11	2.25	232.15	2.74	0.434	0.01	n.i.
H	emotional undereating	2.45	255.09	2.40	235.89	2.80	269.31	3.20	335.97	17.11	0.001	0.03	H.I < H.IV * H.II < H.IV ** H.III < H.IV *
I	food fussiness	2.20	264.30	2.00	255.34	2.00	262.15	2.20	263.76	0.32	0.955	0.00	n.i.
J	slowness in eating	2.88	277.13	2.50	232.16	2.75	270.46	3.00	308.77	13.41	0.004	0.03	J.II < J.IV *

* *p* < 0.05, ** *p* < 0.01. Mdn: median, Mrang: mean range, H (3): H Kruskal–Wallis test, *p*: *p*-value,ε^2^: measurement of the strength of the effect by the epsilon2 coefficient.

## Data Availability

The data that support the findings of this study are available on request from the corresponding author, J.B.

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
