# Peer review of "Eating Behaviours of Polish and Portuguese Adults—Cross-Sectional Surveys"

_nutrients, 2023, doi:10.3390/nu15081934_

Round 1

Reviewer 1 Report

Underweight, BMI score < 18.5 kg/m2 (not ˃)

Author Response

Dear Reviewer,

thank you for your precious time. We are grateful for a very kind review. 

Regarding comment "Underweight, BMI score < 18.5 kg/m2 (not ˃)", manuscript has been corrected according to Reviewer suggestions.

Reviewer 2 Report

There is no main objective of development work in this paper.

Review the summary. Has repetitive phrase.

Author Response

Dear Reviewer,

thank you for your precious time and kind review. 

Manuscript has been corrected or clarified according to Reviewer suggestions and marked red in the provided manuscript. Also, all manuscript has been checked by a native speaker - spelling changes as suggested by Editor’s are in “Track changes” function.

Reviewer 3 Report

Interesting paper on knowing and comparing the eating behaviors of Polish and Portuguese adults and, in addition, relating daily behaviors, eating approach and food avoidance behaviors and BMI in both groups.

Appropriate methodology for the defined objectives and appropriate statistical analysis of the collected data.

Discussion and conclusions based on the results obtained.

Author Response

Dear Reviewer,

thank you for your precious time. We are grateful for a very kind review. 

Reviewer 4 Report

Dear Autor’s

Some corrections and adaptation that we suggest for the publication, nothing that you are not able to do but that are necessary to do.

Line 104 the reference 15 is for polish study, and do not reflect Portuguese diet.

Line 135 to 137 you write: “The average daily calorie intake for a person is significantly high, between 3,480 (data from 2018) - 4,075 (data from 2016-2020, including first period of the pandemic) calories, which is twice the recommended value for an adult average, healthy weight. [23-25].” Those data are not average daily calorie intake, but they are measures of food consumption from the point of view of food supply. It is expressed in daily supplies per habitant, it is not the same thing! When you write that it is twice the recommended value for adult average you are compared two different things: average daily calories recommendation versus food supply, it is not correct.

Line 173 you write “In Poland, the average calorie intake in 2018” it is the same problem it is not calorie intake, it is food supply.

Materials and methods: please put a reference for the AEBQ, specify with reference the adapted version used for Poland and for Portugal. The number of men’s and women’s in each population must be specified.

Table 1. the signification of R M SD Mdn Sk Kurt D must be in the bottom of the table, D (distribution) significance of what and witch statistical test must also be clarified in this place.

Line 239 to 241 you write: Study's findings also showed that compared to the Portuguese, the Poles were older, taller and heavier. Detailed descriptive statistics for the described sample can be found in Table 1. Where is the statistic of those results presented? The number of men’s and women’s is in the same proportion in both populations? This must be clarified because it may influence the height, weight and BMI of the populations and can be a reason for the differences or not, but we have to know to interpret.

 Table 2, 5, 6, 7, 8, 9, 10. at the bottom signification of the abbreviations and the statistical test used must be indicated

Table 3, 5, 8, 10, 11 From where does come from the data of those tables, where is it explained in materials and methods, does it exist validated versions for Portuguese and Polish? What are the references? All the questionnaire and questions used in this study must be clarified in material and methods with references for the originals, and the translation validations

Table 4. from witch questionnaire come from those data’s? What about translation?

Author Response

Dear Reviewer,

Thank you very much for the valuable comments, which we took into account when correcting the manuscript. All changes are in “Tracked changes” function and marked in red in the attached manuscript.

Below, we would like to present answers to comments.

Line 104 the reference 15 is for Polish study, and does not reflect Portuguese diet.

Line 104 the reference has been changed, thank you for your suggestion. 

Line 135 to 137 you write: “The average daily calorie intake for a person is significantly high, between 3,480 (data from 2018) - 4,075 (data from 2016-2020, including first period of the pandemic) calories, which is twice the recommended value for an adult average, healthy weight. [23-25].” Those data are not average daily calorie intake, but they are measures of food consumption from the point of view of food supply. It is expressed in daily supplies per habitant, it is not the same thing! When you write that it is twice the recommended value for adult average you are compared two different things: average daily calories recommendation versus food supply, it is not correct. 

Line 173 you write “In Poland, the average calorie intake in 2018” it is the same problem, it is not calorie intake, it is food supply. 

Line 135-137 and Line 173. Thank you for your comment. It has been corrected according to Reviewer suggestions.

Materials and methods: please put a reference for the AEBQ, specify with reference the adapted version used for Poland and for Portugal. The number of men’s and women’s in each population must be specified.

Materials and methods - reference has been added. The information about validation of AEBQ has been added to both, Polish and Portuguese sample.

Number of women and men in both groups have been added. 

Table 1. the signification of R M SD Mdn Sk Kurt D must be in the bottom of the table, D (distribution) significance of what and witch statistical test must also be clarified in this place. 

Table 1. Manuscript has been corrected according to Reviewer suggestions.

Line 239 to 241 you write: Study's findings also showed that compared to the Portuguese, the Poles were older, taller and heavier. Detailed descriptive statistics for the described sample can be found in Table 1. Where is the statistic of those results presented? The number of men’s and women’s is in the same proportion in both populations? This must be clarified because it may influence the height, weight and BMI of the populations and can be a reason for the differences or not, but we have to know to interpret. 

Line 239 to 241: The new table (Table 2) has been added regarding this issue.

Table 2, 5, 6, 7, 8, 9, 10. at the bottom signification of the abbreviations and the statistical test used must be indicated

Table 3,6,7,8,9,10,11 (new numbers after adding Table 2) - abbreviations and tests have been added as suggested. 

Table 3, 5, 8, 10, 11 From where does come from the data of those tables, where is it explained in materials and methods, does it exist validated versions for Portuguese and Polish? What are the references? All the questionnaire and questions used in this study must be clarified in material and methods with references for the originals, and the translation validations.

Table 4,6,9,11,12 (new numbers after adding Table 2) - Those questions were mentioned in Material and Methods as: „Individuals also answered questions about perception of current body image and daily behaviours in reference to consumed meals”.  Questions were based on FFQ questionnaire that was validated for Portuguese sample in previous studies but due to different data needed to this study we decided not to use full original version of FFQ. We self-made questions and validated them with group of 15 Portuguese and 15 Polish people before the start of the study to be sure of the clarity of the questions.

Table 4. from witch questionnaire come from those data’s? What about translation?

Table 5 (previous Table 4) - Obtained data are from self-made questionnaire. Those questions were mentioned in Material and Methods as: „Individuals also answered questions about perception of current body image and daily behaviours in reference to consumed meals”.  

Round 2

Reviewer 4 Report

Dear Authors,

Thank you for your corrections

Last concern, you write in you answer to me: “Those questions were mentioned in Material and Methods as: „Individuals also answered questions about perception of current body image and daily behaviours in reference to consumed meals”. Questions were based on FFQ questionnaire that was validated for Portuguese sample in previous studies but due to different data needed to this study we decided not to use full original version of FFQ. We self-made questions and validated them with group of 15 Portuguese and 15 Polish people before the start of the study to be sure of the clarity of the questions.” We understand perfectly you answer, that you must include in your methodology with the references of the FFQ for Portugal and Poland, like that every body know from were does come from the question you used, and that you made an adapted questionnaire with validation.

We wish you success in the publication of your article

Author Response

Dear Reviewer,

thank you one more time for the valuable comment and explanation. We added this part to Materials and Methods section in Manuscript.

Changes are in “Tracked changes” function and marked in red in the attached manuscript.

Thank you for your precious time.